# The Role of Intrinsic and Extrinsic Sensory Factors in Sweetness Perception of Food and Beverages: A Review

**DOI:** 10.3390/foods8060211

**Published:** 2019-06-14

**Authors:** Qian Janice Wang, Line Ahm Mielby, Jonas Yde Junge, Anne Sjoerup Bertelsen, Ulla Kidmose, Charles Spence, Derek Victor Byrne

**Affiliations:** 1Department of Food Science, Faculty of Science and Technology, Aarhus University, 5792 Aarslev, Denmark; lineh.mielby@food.au.dk (L.A.M.); jonas.junge@food.au.dk (J.Y.J.); annesbertelsen@food.au.dk (A.S.B.); ulla.kidmose@food.au.dk (U.K.); derekv.byrne@food.au.dk (D.V.B.); 2Crossmodal Research Laboratory, Department of Experimental Psychology, University of Oxford, Oxford OX2 6GG, UK; charles.spence@psy.ox.ac.uk

**Keywords:** sugar reduction, multisensory integration, intrinsic factors, extrinsic factors, sweetness perception

## Abstract

When it comes to eating and drinking, multiple factors from diverse sensory modalities have been shown to influence multisensory flavour perception and liking. These factors have heretofore been strictly divided into either those that are intrinsic to the food itself (e.g., food colour, aroma, texture), or those that are extrinsic to it (e.g., related to the packaging, receptacle or external environment). Given the obvious public health need for sugar reduction, the present review aims to compare the relative influences of product-intrinsic and product-extrinsic factors on the perception of sweetness. Evidence of intrinsic and extrinsic sensory influences on sweetness are reviewed. Thereafter, we take a cognitive neuroscience perspective and evaluate how differences may occur in the way that food-intrinsic and extrinsic information become integrated with sweetness perception. Based on recent neuroscientific evidence, we propose a new framework of multisensory flavour integration focusing not on the food-intrinsic/extrinsic divide, but rather on whether the sensory information is perceived to originate from within or outside the body. This framework leads to a discussion on the combinability of intrinsic and extrinsic influences, where we refer to some existing examples and address potential theoretical limitations. To conclude, we provide recommendations to those in the food industry and propose directions for future research relating to the need for long-term studies and understanding of individual differences.

## 1. Introduction

Eating and drinking are amongst the most multisensory of the experiences that we have. When people think about the consumption of food and drink, the senses of taste and smell usually come to mind first. However, a growing body of research conducted over the last decade or two has increasingly demonstrated that *all* of our senses play a role in influencing flavour perception (see References [1,2,3] for reviews). For instance, recalling the experience of eating an apple will usually evoke not just taste and smell, but also its colour, weight, shape, its firmness, crunchiness, juiciness and even the sound of chewing and perhaps its provenance (e.g., supermarket, organic, local, or the tree in the backyard). 

A large body of research now supports the view that both *food-intrinsic* sensory factors (e.g., product colour, aroma, texture, viscosity, etc.) as well as *food-extrinsic* factors (e.g., visual, olfactory, and tactile properties of product packaging or servingware, background music, ambient lighting, temperature and aroma, etc.) play a role in determining whether we accept and how we perceive food and beverages (e.g., for intrinsic factors [2,4,5] and for extrinsic factors [6,7,8,9,10,11,12]). What is less clear, however, is how these different factors interact and the relative importance of intrinsic and extrinsic factors to our perception of, not to mention our behaviour towards, food and drink. 

In this review, we focus on how intrinsic and extrinsic factors can enhance the perception of sweetness in foods and beverages and address the question of how (and if) they can be combined in order to deliver an enhanced perception of sweetness. The decision to target the perception of sweetness is informed by the growing public health concern over excessive sugar consumption. The consumption of sweet foods has been argued to be one of the major contributors to the current obesity epidemic, with more than 3 million deaths globally each year [13,14,15,16]. Moreover, sugar reduction is of critical concern to major food and beverage companies such as PepsiCo, Givaudan, and Arla, who have been engaging in a number of major initiatives in order to reduce added sugars and develop naturally resourced sweeteners [17,18,19]. Therefore, a multisensory, psychological model of sweetness perception is especially important when it comes to the design of sugar-reduced/replaced foods and beverages. 

Hutchings et al. [20] recently outlined four general strategies for sugar reduction. Sugar substitution, altering food structure (e.g., heterogeneously distributing sucrose, modifying tastant release, or reducing particle size), gradual long-term sugar reduction, and using the principles of multisensory integration. However, Hutchings et al. [20] do not address the role of product-extrinsic factors in sweetness perception. Therefore, in the current review, we endeavour to assess the complex interplay between various different sensory factors surrounding the multisensory eating experience, with a cognitive neuroscience view of how the senses combine to shape taste perception (Note that the scope of the current review is limited to the influence of *sensory cues* on sweetness enhancement. Various cognitive factors, such as, for instance, any information provided by packaging, can also influence consumer flavour perception and acceptance of sugar-reduced foods [21]; however, such cognitive effects fall beyond the scope of the present article. Similarly, sugar substitution schemes with non-caloric sweeteners are also outside of the scope of the present review [22]).

## 2. Food-Intrinsic versus Food-Extrinsic Influences on Sweetness Perception

In the following section, we will target each sensory modality in turn and review the literature on the intrinsic and/or extrinsic cues regarding their influence on sweetness perception. Table 1 provides a representative summary of studies demonstrating sweetness enhancement effects from the influence of different sensory modalities.

### 2.1. Vision

#### 2.1.1. Colour

A growing body of scientific research shows that people systematically associate different colours of foods and beverages (regardless of whether they are found in the food itself or in the food presentation/packaging), with specific basic tastes (see References [49,50,51] for reviews). In one early study, O’Mahony [52] reported that North American participants consistently matched the colour red to sweet taste, yellow to sour tastes, and white to salty tastes. The impact of particular colours on the perception of specific tastes has been repeatedly demonstrated over the years. Specifically, in terms of sweetness, red-coloured drinks have been found to enhance sweetness detection [34], expectations of sweetness [53], and perceived intensity [30,33,35,39,50]. However, in terms of the sensitivity to sweet taste, Maga [36] did not observe any effect of the colour red on taste detection thresholds. Rather, this colour was found to decrease bitter taste sensitivity (see Reference [49] for a review).

What is perhaps more surprising, is the role of extrinsic colour cues—that are not themselves part of the product itself—on perceived taste. So, for instance, Harrar et al. [54] reported that salty popcorn was perceived as slightly, but significantly, sweeter when served in a red or blue bowl, as compared to when presented in a white bowl. Meanwhile, in another study, a pinkish-red strawberry mousse was rated as tasting roughly 15% sweeter when served on a white plate than when served on a black plate [8,55]. Harrar and Spence [56] replicated these results using black versus white spoons with different coloured yoghurts. Furthermore, they posited that the contrast between the colour of the cutlery and the colour of the food may help to explain differences in consumer liking and perceived value of the food [57], even if the effect on sweetness was based on the colour of the cutlery alone. Additionally, it may be the combination of colours that especially signal sweetness; Woods et al [58] examined how single colours compared to colour combinations were rated in terms of sweetness, and found that the combination of pink and white was rated significantly sweeter than white and pink represented individually as single colours. 

When it comes to beverages, hot chocolate was rated as slightly sweeter when served in a dark cream coloured cup as compared to orange, red, or white cups [59]. Elsewhere, it has been demonstrated that the same café latte tasted less sweet in a transparent glass cup with a white sleeve as compared to cups having a blue sleeve or no sleeve at all [60]. Furthermore, the same espresso coffee is both expected to be sweeter in a pink cup compared to a white one and, in fact, does taste sweeter too [29] (also see Reference [61] for a review of the literature on background colour).

#### 2.1.2. Food Shape

Needless to say, shape is not solely a visual attribute but also related to touch, as discussed below. People associate round shapes with sweetness, whereas angular shapes tend to be associated with sourness, saltiness, and bitterness instead [62,63,64,65]. This association has been observed in food shapes [41,66]. That is, food shape can influence expectations. For instance, Wang et al. [41] demonstrated that people expected round-shaped chocolates to taste sweeter and less bitter than angular-shaped chocolates. Alternatively, it is also possible that food with a round shapes feel smoother and creamier in the mouth, and creaminess may be associated with sweetness [24].

Now, beyond the association with abstract shapes and taste, there is also a more semantic association. In a recent study, Spence, Corujo, and Youssef [67] demonstrated that a candy-floss like shape/texture was strongly associated with sweetness in both British and Spanish participants. Like food shape, food-extrinsic shape can also evoke taste similar associations when it comes to plate shape [31,57] (though see Reference [8]) and even packaging shape [68] and typeface/font [69] (see Reference [11] for a review of the typeface of taste).

### 2.2. Audition

#### 2.2.1. Intrinsic-Food Sounds

The sounds of mastication (i.e., the sounds that we hear when eating) can contribute to our perception of crispness, freshness and pleasantness for foods such as crisps, biscuits and fruit [70,71,72,73] (see Reference [74] for a review). For instance, an apple is rated as less crispy and softer when the biting sound is attenuated in either just the high-frequency range, or over the entire sound envelope [70]. However, as far as we are aware, there has not yet been a study published that has investigated whether changing mastication sounds influences sweetness perception.

#### 2.2.2. Extrinsic-Food Sounds

Beyond the sound of eating, incidental sounds playing in the background can also influence our taste perception. Woods et al. [75] conducted studies assessing the impact of loud versus quiet background white noise (75–85 dB versus 45–55 dB) on the perception of sweetness, sourness and liking for a variety of foods. Sweet foods (biscuits and flapjacks) were rated as tasting significantly less sweet in the loud background noise condition compared to the quiet background noise condition. As far as sweetness is concerned, Yan and Dando [76] found similar results when they assessed intensity ratings for a variety of pure taste solutions in silence and with simulated airplane noise (presented at 80–85 dB over headphones). Once again, sweetness ratings were suppressed in the noise condition compared to silence. 

“Sonic seasoning”, the idea that certain auditory stimuli can be deliberately used to alter people’s taste perception, is becoming an increasingly popular topic in both the academic literature as well as the popular press [9,74,77,78,79]. Previously, it has been shown that specific auditory attributes are associated with basic tastes, both when presented as taste words (for instance, “sweet”), and in the form of tasting solutions [80,81,82,83,84,85,86,87,88] (see Reference [89] for a review). For instance, both sweet and sour tastes are mapped to high pitch, whereas bitterness is mapped to low pitch [81,85,86,87]. Crisinel et al. [7] first demonstrated that beyond any cross-modal associations between sounds and taste words, auditory stimuli could also affect people’s taste evaluations. The participants in their study were given samples of bittersweet cinder toffee to evaluate while listening to one of two soundtracks that had been specifically composed to correspond to either sweet or bitter tastes. Crucially, the participants rated the cinder toffee samples higher on the sweet–bitter scale (i.e., sweeter and less bitter) while listening to the sweet soundtrack than while listening to the bitter soundtrack. Similar sonic seasoning effects, specifically involving sweetness, have since been found in a range of food and beverage stimuli ranging from juice to beer, and from chocolate to wine [9,24,25,27,90,91]. Moreover, technology is beginning to be incorporated into smart food packaging, such as the “sonic sweetener” prototype coffee container which can deliver sweet- or bitter-sounding music as people consume the beverage inside [92]. 

### 2.3. Olfaction

#### 2.3.1. Retronasal Olfaction

In terms of the effect of aroma on sweetness perception, the body of research supporting the modifying effects of aromas on sweetness perception is certainly not new. Aromas such as caramel, vanilla, and berry flavours have been found to increase the perception of sweetness, at least in Western participants (e.g., [2,44,45,93,94,95,96,97,98,99,100], see References [1,101] for reviews). Here, it is important to note that previous co-exposure is key to taste enhancement, which can differ with participants from different cultural backgrounds. For instance, in one study, the aroma of vanilla was found to enhance perceived sweetness in French participants more than in Vietnamese participants, but the reverse was true for lemon aroma [101,102]. Furthermore, Frank and Byram [44] studied the perception of sweetness and saltiness in different food matrices: Sucrose-sweetened whipped cream and strawberry aroma, sucrose-sweetened whipped cream and peanut butter aroma, sodium chloride salted whipped cream and strawberry aroma and finally sucrose-sweetened whipped cream and strawberry aroma evaluated with the nose pinched. The strawberry aroma tended to enhance the perception of sweetness; the results also showed that an aroma’s ability to enhance sweetness was aroma-dependent, that an aroma’s ability to enhance taste was tastant-dependent, and that the influence of the strawberry aroma on sweetness was olfactory rather than gustatory in origin.

Attempts have been made to predict the level of sweetness enhancement from different aromas. Schifferstein and Verlegh [45] investigated the effect of congruency in the form of co-occurring aroma-taste pairs. They studied strawberry, lemon and ham aromas together with sucrose and thereby created different aroma-tastant pairs varying in levels of congruency. As expected, they found that strawberry and lemon aromas increased sweetness ratings, but that the aroma of ham did not. However, the ratings of congruency did not contribute to the predictive power of the regression predicting sweetness enhancement. This indicates that congruency is a necessary condition for aroma-induced sweetness enhancement to occur, but that the degree of congruency does not relate to the level of sweetness enhancement. The degree of congruency did, however, affect the pleasantness of the mixtures, but as for congruency, the pleasantness scores could not predict the level of sweetness enhancement.

Stevenson et al. [103] investigated how to best predict aroma-induced sweetness enhancement by screening 12 different aromas, including aromas that smelled “sweet”, “acidic”, or non-food-like. They found that the degree to which an aroma smelled sweet was the best predictor of an aroma’s ability to enhance sweetness of sucrose, but that the degree to which an aroma was perceived to smell sweet was significantly correlated to whether the aroma was regarded to be food or non-food-like. The more an aroma was judged to be food-like, the sweeter-smelling it was perceived. 

Besides being aroma- and tastant-dependent, aroma–sweetness interactions are also dependent on the context. Indeed, comparing samples to a sucrose reference have been shown to not only reduce, but also change the cross-modal effect of aromas on sweet taste (unpublished data). Another important fact to keep in mind is the possible interaction between aroma and sweetener. Several studies have investigated the sweetness-enhancing effect of aromas at varying sucrose levels. While not all studies have shown bigger effects at lower rather than higher sucrose concentrations [44], most studies have [45,104,105]. Indeed, a greater degree of sensory integration has been shown for weaker compared to stronger stimuli [106].

#### 2.3.2. Orthonasal Olfaction

In the majority of studies investigating specific odour-induced changes in taste perception including that of sweetness, odorants and tastants have been presented together intrinsically in liquid mixtures which have been sipped (see References [107,108] for reviews). This means that it has not been possible to study the origin of this effect—in other words, whether it is peripheral or central. However, in a study on aspartame solutions and vanilla aroma, Sakai et al. [109] demonstrated that independent of whether vanilla and aspartame were mixed or delivered separately, sweetness enhancement occurred. These early results therefore suggested, surprisingly, that there is little functional difference between retro- and orthonasal-olfaction when it comes to sweetness enhancement. In a study of strawberry and soy sauce odours on perceived sweetness and saltiness, where odorants and tastants were delivered separately, Djordjevic et al. [110] found taste–smell interactions for congruent odour–taste pairs, thus leading them to conclude that odour-induced changes in taste perception are centrally mediated.

In a study using saccharin solutions and benzaldehyde, Pfeiffer et al. [111] found that out of a panel of 16 people, 12 showed perceptual integration while the remaining four did not. This suggested that there are individual differences in how odour-induced changes in taste perception occurs. While evaluating the intensity of fruit flavour, Hort and Hollowood [112] further found that a subgroup of trained assessors was unaffected by cross-modal aroma-sweetness interactions. Additionally, Pfeiffer et al. [111] found that while retro- and the orthonasal-presentation both produced perceptual integration, the threshold values for the different presentation methods were different.

### 2.4. Taste

Taste–taste interactions are not well understood, partly since research has reported contradictory results. In their review on binary taste interactions at suprathreshold levels, Keast and Breslin [113] suggested that the position of the individual taste stimulus on the concentration-intensity psychophysical curve (expansive, linear, or the compressive phase of the curve) can predict important interactions of taste mixtures, and that these interactions are thus concentration-dependent.

In relation to the perception of sweetness, many compounds can elicit a sweet taste. As sugar reduction is a highly interesting topic for industry, academia, as well as for society in general, several studies have investigated sweet tasting compounds and perceptual interactions between them [114,115,116,117,118]. For instance, the perception of sweetness in mixtures of aspartame and acesulfame K has been extensively investigated and the two sweeteners have demonstrated a perceptual synergy when mixed [32,114,115]. In general, Keast and Breslin [113] suggested that, at low intensity/concentrations (corresponding to the expansive phase of the psychophysical curve), synergy is reported when mixing sweet tasting compounds. On the other hand, at higher intensities/concentrations (the linear of compressive phase of the psychophysical curve) sweetness enhancement is less common and suppression is reported instead. 

The perception of sweetness is, however, also modified by other taste qualities. In their review, Keast and Breslin [113] generalised on the effect of other taste qualities as follows: At low intensities/concentrations (the expansive phase of the psychophysical curve), the perception of sweetness is increased by salty taste, while results on the effect of adding bitterness and sourness was inconclusive. At medium intensities/concentrations (the linear phase), sweetness was suppressed by bitterness while results were inconclusive for saltiness and sourness. At high intensities/concentrations (the compressive phase), both bitterness and sourness suppressed the perception of sweetness while saltiness either suppressed or else had no effect. Similar results were found in the review on heterogeneous binary taste interactions by Wilkie and Capaldi Phillips [119]. However, they found that salt generally suppresses sweet taste, but that, as with Keast and Breslin [113], the effect of salt was concentration-dependent. Only a few studies have investigated the interaction between sweetness and umami, and the results are, thus far, inconclusive [113,119].

With respect to more complex taste mixtures such as ternary and quaternary mixtures including sweetness, Green et al. [120] followed a sip-and-spit procedure to investigate if asymmetries in suppression between stimuli in binary mixtures could predict taste perception in ternary and quaternary mixtures. The authors found a consistent pattern of mixture suppression in which sucrose sweetness was both the least suppressed quality as well as the strongest suppressor of other tastes (sourness, saltiness, and bitterness). Further, they concluded that, the overall intensity of mixtures was found to be predicted best by perceptual additivity. In other words, the overall taste intensity was derived from the sum of the tastes perceived within a mixture, rather than the sum of the perceived intensities of the individual stimuli (stimulus additivity).

### 2.5. Touch

#### 2.5.1. Oral-Somatosensation

The oral texture or mouthfeel of food and drink can influence the way in which multisensory flavours are experienced (e.g., [121,122,123]). Christensen [124] first suggested that increased viscosity can reduce taste perception in sweet and salty solutions. In a similar study, this time involving viscosity and flavour intensity, Bult et al. [121] presented a creamy odour either orthonasally or retronasally using an olfactomer. At the same time, milk-like solutions with different viscosities were delivered to the mouth of the participants. The intensity of perceived flavour decreased as the viscosity of the liquid increased, regardless of whether the odour was presented orthonasally or retronasally. Similarly, Weel et al. [123] studied the perception of ethylbutyrate (pineapple like, fruity) and diacetyl (buttery) flavours in whey protein gels, detected by 10 trained panellists. They found that perceived flavour intensity decreased linearly with increasing gel hardness. 

Of course, it should be kept in mind that flavour compounds can interact with various ingredients in the food matrix through either chemical or physical interactions, possibly leading to either increased or decreased flavour release (e.g., [125,126,127]). However, Tournier et al. [100] found that physico-chemical interactions could not explain all of the sensory interactions that they observed, suggesting that cognitive mechanisms were involved after all. 

Beyond mouthfeel, the temperature of food in the mouth can also influence the perception of taste. In general, the threshold for detection of sweet tastes (as well as bitter, salty and sour) shows a U-shaped response as a function of temperature, with the lowest threshold, i.e., highest sensitivity, in the 20–30 °C temperature range [128,129,130]. Furthermore, some individuals experience a phantom taste when small areas of the tongue are rapidly heated or cooled, with most such thermal tastes experiencing sweetness when the anterior edge of the tongue is warmed [131]. Therefore, it is possible that for these temperature tasting individuals, sweetness may be enhanced when they consume warm foods. 

#### 2.5.2. Tactile Feedback

A recent body of empirical research has demonstrated that the taste and hedonic evaluation of food and beverages can indeed be influenced by the surface texture of packaging materials [59,132], servingware [46,47] and even the food itself [133]. For instance, Biggs et al. [46] served caramelised biscuits on two plates of the same shape, one with a rough and grainy surface texture, the other with a smooth and shiny texture instead. Biscuits taken from the smoother of the two plates were rated as sweeter and less crunchy than those from the rough plate. In another study examining larger-scale surface textures, van Rompay and colleagues [47] demonstrated that 3D-printed surface patterns influenced, amongst other attributes, the sweetness intensity of beverages. Namely, hot chocolate and coffee tasted from a cup with a rounded macrogeometric outer surface pattern were rated as tasting sweeter and less intense than the same beverages when tasted from a cup that had a much more angular outer pattern instead. Conversely, the angular surface pattern increased perceived bitterness and taste intensity when compared to the rounded pattern. Slocombe et al. [133] examined the influence of both food-intrinsic and extrinsic surface texture on sweetness, bitterness, and sourness of citrus-flavoured fondant sugar. While these researchers observed no influence of plate texture (rough versus smooth) on the taste of these food samples, they did observe that, when it came to the surface texture of the food itself, samples were rated as tasting significantly sourer when it had a rough surface texture as compared to a smooth one. Recently, Wang and Spence [48] demonstrated for the first time that haptic feedback of surface texture can apply to orthonasal olfaction as well as to the taste and hedonics of foodstuffs. The authors found that haptic sensations from touching different surface textures (velvet versus sandpaper) can (at least when attended) influence the wine-tasting experience, both in terms of olfaction and in terms of in-mouth sensations. The wine was rated as smelling fruitier, as tasting sweeter and more pleasant, when the participants touched the velvet as compared to when they touched the sandpaper.

## 3. A Neuroscientific Perspective on Sensory Interactions

### 3.1. The Role of Multisensory Flavour Perception

When it comes to rationalising multisensory integration, Gibson [134] proposed an ecological model whereby information about an object is processed and interpreted via different sensory channels, as part of an active process to acquire information about the environment (see Reference [1] for a review). Flavour perception, then, can be considered as a system that controls ingestion, with the goal of picking up all available information about the food that is about to enter the body in order to secure an adequate supply of nutrients and avoid poisons [135]. Moreover, this process can be considered in multiple stages: first, there is the pre-ingestion period when food is identified and expectations are formed—this is probably most naturally gathered via visual information, together with some degree of tactile (e.g., weight, surface texture, hardness), orthonasal olfactory, and auditory information (e.g., sizzling, fizzing, bubbling). Then, there is the actual eating/mastication period where additional properties of the food—such as its taste, retronasal aroma, texture, temperature and piquancy—are detected by various taste and oral-somatosensory receptors. These receptors serve to detect nutrients and poisons in the food [136,137]. At the same time, hedonic judgments are made continuously during ingestion as a way of motivating and curtailing ingestion (e.g., [138]). Finally, learned associations are formed between different sensory stimuli as a result of the eating process (e.g., many red-coloured fruits are ripe and sweet [49]).

Just as the tactile system combines disparate information from various parts of the body and various different classes of receptors to register invariant stimuli, this proposed flavour system combines information from all the senses in order to form flavour percepts that ultimately optimise nutrient intake. Viewed from this perspective, extrinsic information such as packaging colour or background sound can act to provide extra information about the food that one is about to taste or is currently tasting. According to Bayesian decision theory, the brain uses prior knowledge about what sensory signals go together—whether inborn or explicitly learned—to integrate appropriate sensory stimuli with the goal of maximising the reliability of perceived information [139,140,141] and, presumably, to reduce cognitive load by combining disparate sensory cues into a single object. Cross-modal correspondences involving sweetness (such as with round shapes or consonant harmonies), could act as a conduit (i.e., in the form of Bayesian priors) to help the brain interpret multisensory cues in order to help form taste/flavour evaluations.

### 3.2. Evidence of Multisensory Flavour Perception in the Brain

In humans, taste is first projected from the tongue and oral cavity to the primary taste cortex in an area of the anterior insula and frontal operculum (see References [142,143] for reviews), along with oral texture and temperature [144,145].

#### 3.2.1. Aroma + Taste Binding

In support of the idea that the different channels of information contributing to flavour are bound together in a unified entity, neuroimaging evidence has demonstrated that convergence for taste and odour occur in the orbitofrontal and anterior cingulate cortex [146,147,148]. Indeed, increased blood oxygen level dependent (BOLD) response was observed in the orbitofrontal cortex (OFC) and amygdala when taste and odour stimuli are presented in combination, compared to the summed activity of taste and smell when pretended alone [149].

The role of olfaction, however, is made more complicated by the comparison of orthonasal versus retronasal olfaction. Whereas retronasally perceived odours are referred to the oral cavity (see a further discussion on oral referral in Section 4.1), orthonasal odours are perceived to come from outside the oral cavity. For instance, there are differences in neural responses evoked by orthonasal versus retronasal odours, with orthonasal perception leading to preferential activation in the insula, thalamus, hippocampus, amygdala and caudolateral orbitofrontal cortex and retronasal perception leading to preferential activation in the perigenual cingulate and medial orbitofrontal cortex [150]. Interestingly, these different patterns of activation were only found in the odorants associated with food (i.e., chocolate compared to lavender).

#### 3.2.2. Vision + Taste Binding

Vision is crucial in the prediction of food flavour. By altering the expectation of flavour, colour can enhance orthonasal olfactory intensity and reduce retronasal intensity (relative to colourless solutions, see Reference [151]). Furthermore, there is evidence to suggest that visual inputs associated with food also converge with olfactory and taste information in the OFC [152,153,154] (see References [155,156] for reviews). Multisensory neurons in the OFC have parallel sensitivities to the input quality from the different modalities; neurons which are most responsive to sweet tastants (glucose) also have a stronger response to the visual representations of sweet fruit juice or olfactory stimuli of fruit odours [157].

#### 3.2.3. Sound + Taste Binding

The influence of sound on taste and flavour perception can be considered in terms of expectations, emotion mediation, attentional direction, physiological reaction or just bias [79,158]. Of course, there is the possibility that any cross-modal effects of music on taste perception may have, at least in part, a direct, low-level, physiological basis (although see Reference [159] for one reported null-effect of a specific sour soundtrack on increasing salivation levels). Such a suggestion is inspired by Wesson and Wilson’s [160] surprising discovery of direct connections between the ear and the olfactory tubercule in mice (see Reference [161] for a review of the multiple functions of the olfactory tubercule). In another example of rodent olfactory–auditory integration, single-cell recordings in the primary auditory cortex of female mice revealed that exposure to a pup’s body odour can reshape neuronal response to pure tones and natural auditory stimuli [162]. It is certainly feasible that such a cross-modal connection might exist in humans, especially as some exciting preliminary neuroimaging results have recently started to appear demonstrating activation in the primary taste cortex by taste-congruent soundtracks [163].

A different possible neural connection was suggested by Brown et al. [164] who demonstrated that the processing of aesthetic stimuli—be they paintings, music or food—overlap within the primary gustatory cortex. The authors propose that the aesthetics system evolved first for the appraisal of objects necessary for survival, such as evaluating the suitability of food/energy sources; over time, according to their suggestion, the same neural circuitry was co-opted for the appreciation of artworks. Therefore, the fact that our evaluation of music and food are processed in overlapping brain areas might potentially account for the associations that we make between them, not to mention how the evaluation (especially hedonic) of stimuli in one sensory modality might influence the evaluation of another.

#### 3.2.4. Touch + Taste Binding

Oral-somatosensory information regarding the food we happen to have in our mouths is transferred to the brain via the trigeminal nerve [165]. The triminal nerve then carries information concerning touch, texture, temperature, proprioception, nociception and chemical irritation from the receptors in the mouth directly to the primary somatic sensory cortex of the brain [166,167]. The importance of somatosensory perception is demonstrated in the large areas of the sensory cortex dedicated to the lips and tongue [168]. Moreover, oral texture (including the perception of fattiness in food) is also represented in the OFC [169,170]. Congruent combinations of colour and orthonasally presented odours have been shown to lead to enhanced activation in the OFC [171]. To the best of our knowledge, neuroimaging techniques have not yet been applied to the case of somatosensory–taste interactions.

## 4. A Framework for How Intrinsic and Extrinsic Factors Influence Multisensory Flavour Perception

### 4.1. Differences between Exteroceptive and Interoceptive Senses

When thinking about the senses and their role in multisensory flavour perception, it can be helpful to distinguish between two categories: the exteroceptive sense of vision, audition, and orthonasal olfaction are typically stimulated prior to (and sometimes during) the consumption of food, and the interoceptive senses (retronasal olfaction, oral-somatosensation and gustation) are those that are stimulated during eating [172]. In the latter case, the relevant senses are taste, retronasal smell, oral-somatosensation and the sounds associated with the consumption of food. Different brain mechanisms may be involved in these two cases. Small et al. [173] found different and overlapping neurological representations of anticipatory and consummatory phases of eating; specifically, the amygdala and mediodorsal thalamus respond preferentially to odours associated with a nutritive drink, whereas the left insula/operculum responds preferentially to the consumption of the drink itself. The right insula/operculum and left OFC responded preferentially to both anticipatory and consumptive phases. Overall, it would seem likely that the multisensory integration of interoceptive flavour cues is more automatic than the combination of cues that is involved in interpreting exteroceptive food-related signals [1,174,175]. 

One of the most important means by which exteroceptive cues influence food perception relates to expectancy effects [176,177,178,179]. That is, visual appearance cues, orthonasal olfactory cues, and distal food sounds can all set up powerful expectations regarding the food that someone is about to eat. When the food or drink is then evaluated, assimilation may occur if there is only a small discrepancy between what was expected and what was provided. However, if the discrepancy between expectations and the actual interoceptive information is too large, then contrast may occur instead. Human neuroimaging and animal electrophysiology has shown that expectations can modulate sensory processing at both early and late stages, and the response modulation can be either dampened or enhanced (see References [180,181,182] for reviews). 

Another example of differences between interoceptive and exteroceptive senses come from Koza et al. [151]. These researchers demonstrated that colour had a qualitatively different effect on the perception of orthonasally (interoceptive) versus retronasally (exteroceptive) presented odours associated with a commercial fruit-flavoured water drink (see also References [124,183]). In particular, they found that colouring the solutions red led to odour enhancement in those participants who sniffed the odour orthonasally, while leading to a reduction in perceived odour intensity when it was presented retronasally. The authors suggested that this surprising result may be accounted for by the fact that it may be more important for us to correctly evaluate foods once they have entered our mouths, since that is when they pose a greater risk of poisoning. By contrast, the threat of poisoning from foodstuffs located outside the mouth is less severe. Alternatively, however, it may well be that people simply attend more to the stimuli within their bodies as compared to those stimuli that are situated externally [141], and that this influence biased the pattern of sensory dominance that was reported.

Given the above considerations, rather than a food-intrinsic versus food-extrinsic divide, it may be more appropriate, with neuroscience and physiology in mind, to divide sensory cues depending on where it is referred. In other words, the key question to consider here is, is the sensory stimulus localised (or perceived to be) coming from within or outside the mouth?

### 4.2. Oral Referral

The importance of the oral cavity can be seen through the observation that flavours appear to originate from the oral cavity, even if olfactory stimuli are detected in the nose (e.g., [184,185,186], see Reference [187] for a review). In addition, the phenomenon of oral referral appears to go beyond merely changing the perceived location of olfactory stimuli; in fact, they are combined with taste information from the tongue to form integrated flavour percepts that cannot be attended to separately [184,188]. Notably, people find it difficult to attend selectively to olfactory stimuli after the stimuli have been localised in the mouth [188,189]. The loss of the source of olfactory information is most likely a result of gustatory attention capture (according to Reference [187]), where the most intense stimulus (normally taste) directs one’s attention to the spatial location where that stimulus comes from. This is supported by studies indicating that the degree of oral referral is proportional to the intensity of the tastants, and inversely proportional to the intensity of olfactory stimuli [186]. 

Intriguingly, the occurrence of oral referral also seems to be related to the degree of congruency between the oral and taste stimuli. Lim and Johnson [190] demonstrated that, when participants were introduced to a simultaneous retronasal odour (soy sauce, vanilla) and a taste solution (sweet, salty, water), they rated the odours as coming from the mouth significantly more often when the odour–taste combination was congruent (vanilla–sweet, soy sauce–salty) than when the solution was neutral or when the combination was incongruent. Further studies conducted with solid gelatine disks instead of liquid solutions [191], and with more ecologically valid stimulus combinations (citral aroma with sweet or sour tastants, coffee aroma with sweet or bitter tastants) revealed similar results where oral referral was enhanced proportional to the degree of self-reported smell–taste congruency [192]. In addition, more recent research supports the hypothesis that retronasal enhancement of odour by taste is dictated by the nutritive value of the tastants in addition to odour–taste congruency; sweet, salt, and umami tastes—which signal the presence of elements essential for survival—presented evidence of enhancing retronasal odour, but no such effect was seen for sour or bitter tastes [193]. In the context of sweetness perception, then, it certainly seems that multisensory cues localised in the mouth (such as food-intrinsic aroma or textural cues) would be more effective in enhancing sweetness perception than those cues localised elsewhere.

## 5. Combining Intrinsic and Extrinsic Influences

There has been relatively little research on the interaction between food-intrinsic and food-extrinsic factors. The available cognitive neuroscience research suggests that the biggest impact on our experiences and behaviours occur when several sensory attributes are changed at once, and when they complement one another [172]. This is precisely the sort of situation in which one might expect to see an additive response (both in the brain and in behaviour), a response that is far bigger than that which can be achieved by manipulating a single sense individually at a time [106,194].

### 5.1. Intrinsic–Intrinsic Interactions

Several studies have examined interactions between intrinsic factors in relation to sweetness. For instance, Zampini and his colleagues [195] found that an orange-coloured water solution only was found significantly sweeter than other colours in combination with orange flavour. Similarly, in several studies, Junge [196] investigated interactions between aroma and colour in relation to sweetness, as well as between aroma and viscosity, in an apple/elderflower fruit drink. In one study conducted with trained panellists, Junge [196] found that both pomegranate aroma and red (different from the original yellowish) colour increased sweetness and showed normal additivity in mixture. In another study [196] conducted with trained panellists, increasing the viscosity of the fruit drink with pectin resulted in a tendency to an increase in rated sweetness. This tendency was maintained even if vanilla and pomegranate aromas were added. On the contrary, Knoop et al. [197] found that both a modification of texture and of aroma had a sweetness enhancing effect on an apple juice gel, but in combination, these showed the same sweetness enhancement as aroma and texture had individually. Thus, the interaction effect was smaller than the sum of the two individually and thereby not additive. 

Besides cross-modal additive effects, it is also worth considering intramodal additive effects between different stimuli. For instance, Woods et al. [58] found that the pairwise combinations of colours had stronger associations with sweetness as compared to single colours. Along a similar line of thought, given the efficacy of sweet-smelling aromas in enhancing perceived sweetness, it would be worth testing whether combinations of sweet smells might have a bigger overall impact compared to just individual smells. 

In terms of interactions between food-intrinsic factors, it has also been demonstrated that taste–aroma interactions are moderated by the nature of the food matrix in question. Labbe et al. [95] tested the taste enhancement effects of cocoa and vanilla flavouring in cocoa and caffeinated milk. They found that in the cocoa beverage, cocoa flavouring led to an enhancement of bitterness and vanilla flavouring enhanced sweetness. However, when it came to the relatively less familiar caffeinated milk product, the addition of vanilla flavouring unexpectedly enhanced bitterness instead of sweetness. Elsewhere, Alcaire et al. [198] found that while an increase in vanilla flavour in a dairy dessert product had a minor effect on sweetness enhancement, the combination of increased vanilla concentration together with higher starch concentration led to an increase in vanilla flavour intensity as well as an increase in perceived sweetness. This was presumably due to the thickened viscosity of the dessert product from the addition of starch.

### 5.2. Extrinsic–Extrinsic Interactions

A few studies have also examined the interaction between various food-extrinsic factors. For instance, Spence et al. [194] conducted a study in which the participants tasted the same red wine in an opaque black glass under different illumination and music conditions. In Experiment 1, the wine was liked more (by approximately 5%) in the combined sweet music and red lighting condition, compared to red lighting alone without music and compared to the control condition (white light, silence). However, there were no changes in rated flavour intensity or freshness (compared to fruitiness). In Experiment 2, the wine was rated as being significantly fresher (an increase of approximately 14%) when tasted under green lighting while listening to sour music, as compared to green lighting alone and to the control condition. What the studies did not assess, however, was the individual effect of music, which leaves the question of whether both visual and auditory cues are required for a noticeable effect on wine perception. 

Elsewhere, Fairhurst et al. [31] manipulated both plate shape (extrinsic) and the arrangement and shape of the food (intrinsic) in a restaurant setting. Participants rated the sweetness, sourness, intensity and pleasantness of two beetroot salads served on either round or angular plates, with the beetroots cut into either angular or round shapes. One group were served beetroot salads only from round plates, and the other group only from angular plates. Serving the salad in the round plate plus round food shape led to significantly sweeter ratings than the congruent angular plate plus angular food shape condition, with the incongruent conditions (where plate shape did not match food shape) rating somewhere in-between.

### 5.3. Intrinsic-Extrinsic Interactions

Recently, Wang et al. [28] conducted a study measuring the relative influence of both extrinsic and intrinsic factors. In a mixed model design, participants tasted samples of apple-elderflower nectars under different visual (looking at a pink or pale-yellow rectangle on an iPad screen) and auditory (listening to bitter or sweet soundtrack) conditions. Three levels of added pomegranate aroma (none, medium level, high level) were also included as an additional between-participant food-intrinsic manipulation. The results revealed that both added aroma and the presence of the soundtrack exhibited significant sweetness enhancement effects. An interaction analysis revealed that the enhancement effect of aroma and soundtrack were independent of each other, and the combined influence of aroma and soundtrack appeared to be greater than the influence of aroma or soundtrack when assessed individually. In other words, there appeared to be a linearly additive relationship between the enhancement effect of aroma and the enhancement effect of soundtrack, even when one factor was intrinsic (aroma) and the other extrinsic (soundtrack). This result gives additional support to the theory that changing multiple sensory stimuli at once could result in a greater sweetness enhancement effect than manipulating single factors alone.

### 5.4. Theoretical Limitations

In terms of sugar reduction, the goal is typically to maintain consumer satisfaction even if the consumers can detect that sweetness has been altered. Intriguingly, while a reduction in added sugar of 20% (from 9% to 7.2% sugar) for chocolate-flavoured milks was detected by trained assessors and consumers, consumer liking was unaffected by the reduction in sugar, for not only 20% but even up to a 40% reduction [105]. Similarly, for orange nectar drinks, lowering added sugar from 10% to 8.5% did not influence sensory attributes or acceptance [199]. In fact, the authors [199] suggested a strategy of long-term gradual reduction in added sugar, from 8.5% to 7.2% and eventually 5.5%. One challenge in working with sugar reduction, as shown in Section 5.1, is that it is food-matrix-dependent, therefore, it is difficult to compare acceptable levels of sugar reduction across the board. 

It is important to keep in mind that exteroceptive sensory cues mostly operate by shaping sensory expectations (see Reference [182] for a review). However, expectancy effects have their limits. Yeomans et al. [200] found that a salmon-based mousse that looked like strawberry ice cream evoked a negative hedonic reaction when participants wrongly assumed it was strawberry ice cream. This shows that there is an envelope in which expectations for a food or drink, such as that driven by food colour, can affect food evaluation via assimilation. Assimilation can only go so far before the disconfirmation of expectations become too great and contrast sets in. The size for such an envelope remains unclear to this day, although it is doubtlessly food and participant-dependent. Table 1 shows that the degree of sweetness enhancement by extrinsic factors (at least, those due to the expectation effects) is usually around 10–20%. 

Finally, it is also important to keep in mind that besides adding a sweet taste to a food or beverage matrix, sugar plays multiple roles in foods and beverages including being a bulking agent as well as contributing to the mouthfeel and texture (see Reference [201] for a review). Therefore, when reducing and/or replacing the sugar, it is necessary to find a substitute which can stand in its place from a structural perspective. However, given the many functionalities sugars have in foods and beverages, reducing and/or replacing it can be a rather complicated process.

## 6. Conclusions, Future Directions, and Open Questions

The results summarised here are of relevance for those working on understanding human sweetness perception, as well as those working to design healthier, sugar-reduced food products. Indeed, the knowledge that multiple sensory cues can, at least under the appropriate conditions, work in conjunction for a greater taste modulation effect will allow designers to come up with more effective sugar-reduced products without taking away consumer satisfaction. It also presents a convincing argument for food researchers and packaging designers to work together, in the philosophy of designing the “Total Product Experience” [202] in order to optimally balance product-intrinsic and extrinsic cues for maximal sweetness enhancement.

### 6.1. Future Directions and Open Questions

#### 6.1.1. Long-Term Studies

As mentioned by Pineli et al. [199], one potential strategy going forward is to gradually lower added sugar % in the long term, in order for consumers to adapt to and eventually accept reduced sugar beverages. Unfortunately, very few long-term studies have been performed in this area. One such study was conducted by Chung and Vickers [203] using sweetened teas. Participants initially tasted three types of tea, each at a low and optimum level of sweetness. Then, over each of the next 20 consumption sessions, participants selected a tea, tasted it, and rated their liking, tiredness, and satisfaction with their choice. The authors observed that liking for the low sweetness levels did increase over time. Moreover, there was a trend for participants to become more tired of the tea at optimum sweetness while becoming less tired of the lower sweetness teas. That said, given the choice of all six teas at each session, participants did not choose the less sweet teas more frequently, as the authors hypothesised. This implies that long-term strategies should focus on improving consumer acceptability for only the reduced sweetness product at any one time, without offering the full-sweetness product as an alternative. As a testament to the difficulty of gradual sugar reduction, while UK retailers have successfully reduced the salt content of breakfast cereals by 47% between 1992 and 2015 via incremental salt reduction targets, sugar content has not changed significantly [204]. The authors speculate that sugar reduction is much more difficult because sugar influences not only the flavour, but also the texture and structure of breakfast cereals. 

#### 6.1.2. Individual Differences

Given that we all live in different taste worlds, it is important to consider whether our taster status influences the extent to which sensory factors influence the way we perceive sweetness. One particularly interesting result to have emerged in this regard comes from Zampini et al. [195]. These researchers found that supertasters showed less visual dominance over their perception of flavour than medium tasters who, in turn, showed less visual dominance than non-tasters. The participants in this particular study had to identify the flavour of a large number of fruit-flavoured drinks presented among other flavour less drinks. The drinks could be coloured red, orange, yellow, grey or else presented as clear and colourless solutions. Overall, the non-tasters correctly identified 19% of the solutions, the medium tasters 31% and the supertasters 67% of the drinks that they were given to taste. Interestingly, the addition of colouring had the largest effect on the performance of the non-tasters, less of an effect on the medium-tasters, and very little effect on the colour identification responses of the supertasters.

Genetic differences in terms of supertaster status have also been demonstrated to play a role in terms of sonic seasoning effects, whereby listening to specific soundtracks congruent to specific tastes/flavours can alter the perception of food. Using a mixed-model design, Wang tested 27 participants who tasted 70% and 85% cacao chocolates while listening to sweet and bitter soundtracks [158]. All participants then took a Phenylthiocarbamide (PTC) taste strip test at the end of the study. Results revealed an intriguing difference when it came to the influence of music. While there were no differences between the two taste sensitivity groups for 70% chocolate, when it came to the more bitter 85% chocolate, the high taste sensitivity group appeared to be more influenced by the different soundtracks than the low sensitivity group (i.e., they found a bigger difference in the taste of the 85% chocolate between the bitter and sweet soundtrack; see [205]).

Moreover, any sugar reduction strategies likely also need to consider individual differences in sweetness liking. Looy et al. [206] categorised people into sweet likers and dislikers based on their hedonic response to sugar solutions of increasing sweetness. Notably, the authors found that the sweetness liker–disliker distinction held across different sugar types (sucrose, glucose, fructose), and even when the solution was coloured red and strawberry-flavoured. Moreover, sweet liking has been demonstrated to be partly inherited [207] and linked with alcoholism [208]. It is important to realise that sweetness-enhancing multisensory cues reviewed earlier may have different outcomes when it comes to consumer acceptance of sugar-reduced foods. For instance, odours which were associated with sweetness via training became more pleasant for sweet likers but more unpleasant in sweet dislikers [209].

### 6.2. Industry Implications

The current review on the role of intrinsic and extrinsic sensory factors in sweetness perception of food and beverages suggests that, changing multiple sensory stimuli, intrinsic as well as extrinsic, at once could result in a greater sweetness enhancement effect than manipulating single factors alone such as working with intrinsic sensory factors only. This has industry implications since in research as well as in academia, work on intrinsic and extrinsic factors are considered as belonging to different disciplines. In industry, this is often reflected in the organizational structure. Namely, Research and Development (R&D), which is in charge of food-intrinsic properties, often sits far away from, and actually has little interaction with the marketing department, who may be responsible for food-extrinsic decisions, such as those involving product packaging. 

However, in order to fully maximize sweetness enhancement in foods and beverages, R&D and the marketing department should work closely together in order to optimally balance product-intrinsic and extrinsic cues. Efforts and resources are unarguably needed to create successful collaborations, but they are crucial to get the full potential of sweetness enhancement. The current review highlights the importance of understanding how food-intrinsic and extrinsic factors work together to form our overall perception of sweetness. Of course, while the topic of this review is focused on sugar reduction, similar strategies can be considered for the reduction of salt/fat content in food to promote healthy eating behaviour. 

## Figures and Tables

**Table 1 foods-08-00211-t001:** A representative selection of studies demonstrating sweetness enhancement via food-intrinsic and extrinsic sensory cues.

Study	Sense	Intrinsic or Extrinsic	Sweet Enhancing Stimuli	Control/Comparison Stimuli	Taste Stimuli	Scale	% Difference
Crisinel et al. (2012) [7]	Hearing	Extrinsic	Sweet soundtrack	Bitter soundtrack	Cinder toffee	1–9 rating (bitter–sweet)	15%
Höchenberger et al. (2018) [23]	Hearing	Extrinsic	Sweet soundtrack	Bitter soundtrack	Toffee	0–100 rating (bitter–sweet)	8%
Höchenberger et al. (2018) [23]	Hearing	Extrinsic	Sweet soundtrack	Bitter soundtrack	Toffee	0–100 rating (sweet, bitter, salt, sour)	No significant difference
Reinoso Carvalho et al. (2016) [9]	Hearing	Extrinsic	Sweet soundtrack	Bitter soundtrack	Belgian beer	1–7 rating sweetness	20%
Reinoso Carvalho et al. (2016) [9]	Hearing	Extrinsic	Sweet soundtrack	Sour soundtrack	Belgian beer	1–7 rating sweetness	20%
Reinoso Carvalho et al. (2017) [24]	Hearing	Extrinsic	Legato soundtrack	Staccato soundtrack	Dark chocolate	1–7 rating sweetness	11%
Wang and Spence, (2016) [25]	Hearing	Extrinsic	Consonant soundtrack	Dissonant soundtrack	Fruit juice (apple, orange, grapefruit)	1–10 rating (sour–sweet)	19%
Wang and Spence (2017) [26]	Hearing	Extrinsic	Consonant soundtrack	Dissonant soundtrack	Fruit juice (apple, orange, grapefruit)	0–10 rating (sour–sweet)	17%
Wang and Spence, (2017) [27]	Hearing	Extrinsic	Sweet soundtrack	Sour soundtrack	Off-dry white wine	0–10 rating sweetness	19%
Wang et al. (2019) [28]	Hearing	Extrinsic	Sweet soundtrack	Bitter soundtrack	Apple elderflower juice	1–9 rating sweetness	8%
Carvalho and Spence (2019) [29]	Sight	Extrinsic	Pink coffee cup	White coffee cup	Espresso	0–10 rating (sweetness)	30%
Clydesdale et al. (1992) [30]	Sight	Intrinsic	More red colouring	Less red colouring	Dry beverage base and sugar solution	1–7 rating sweetness	14%
Fairhurst et al. (2015) [31]	Sight	Both	Round plate and round food presentation	Angular plate and angular food presentation	Beetroot salad	0–10 rating sweetness	17%
Frank et al. (1989) [32]	Sight	Intrinsic	Red colouring	No colour	Sucrose solution	Rating sweetness	No effect
Hidaka and Shimoda (2014) [33]	Sight	Intrinsic	Pink solution	No colouring	Sucrose solution 4% and 6%	10 cm visual analogue scale (VAS) less–sweeter	40%
Johnson and Clydesdale (1982) [34]	Sight	Intrinsic	Darker red coloured solution	Lighter red reference solution	Sucrose solutions 2.7–5.3%	Magnitude estimation sweetness	2–10%
Lavin and Lawless (1998) [35]	Sight	Intrinsic	Darker red solution	Lighter red solution	Fruit beverage + aspartame to 9% sucrose level	1–9 category scale sweetness	10%
Lavin and Lawless (1998) [35]	Sight	Intrinsic	Lighter green solution	Darker green solution	Fruit beverage + aspartame to 9% sucrose level	1–9 category scale sweetness	8%
Maga (1974) [36]	Sight	Intrinsic	Red colouring	Green, yellow, uncoloured solutions	Sucrose solution	Recognition threshold	No effect
Pangborn and Hansen (1963) [37]	Sight	Intrinsic	Red solution	Green, yellow, uncoloured solutions	Pear nectar	Rating sweetness	No effect
Pangborn et al. (1963) [38]	Sight	Intrinsic	Pink colouring	Yellow, brown, light red, dark red colouring	White wine	Rating sweetness	Rose sweetest
Pangborn (1960) [39]	Sight	Intrinsic	Red colouring	Green, yellow, uncoloured solutions	Sucrose solution	2-AFC (alternative forced choice) which one sweeter	No effect
Pangborn (1960) [39]	Sight	Intrinsic	Red colouring	Green, yellow, uncoloured solutions	Pear nectar	2-AFC which one sweeter	No effect
Piqueras–Fiszman et al. (2012) [8]	Sight	Extrinsic	White plate	Black plate	Strawberry mousse	10 cm sweetness scale	15%
Stewart and Goss (2013) [40]	Sight	Extrinsic	White plate	Black plate	Cheesecake	10 cm sweetness scale	28%
Wang and Spence (2017) [26]	Sight	Extrinsic	Image of happy child	Image of sad child	Fruit juice (apple, orange, grapefruit)	0–10 rating (sour–sweet)	20%
Wang et al. (2017) [41]	Sight	Intrinsic	Round shape	Angular shape	Dark chocolate	1–9 rating expected sweetness	30%
Dalton et al. (2000) [42]	Smell	Extrinsic (Orthonasal)	Benzaldehyde odour (cherry almond aroma)	No odour	Saccharin solution	Threshold test	29% increase in benzaldehyde threshold in benz + saccharin condition
Delwiche and Heffelfinger (2005) [43]	Smell	Intrinsic (Retronasal)	Pineapple odour, high concentration	Pineapple odour, lower concentration	Aspartame/acesulfame potassium solution	2-AFC threshold detection	Additive taste-odour
Frank and Byram (1988) [44]	Smell	Intrinsic (Retronasal)	Strawberry odour	No odour	Sweetened whipped cream	0–20 rating sweetness	13% at 0.6 M and 1.2 M; 40% at 0.25 M
Frank et al., 1989 [32]	Smell	Intrinsic (Retronasal)	Strawberry odour	No odour	Sucrose solution	0–20 rating sweetness	~18% at 0.3 M, 7% at 0.5 M concentration
Schifferstein and Verlegh (1996) [45]	Smell	Intrinsic (Retronasal)	Strawberry odour, lemon odour	No odour	Sucrose solution	150 mm sweetness scale	25%
Wang et al. (2019) [28]	Smell	Intrinsic	Pomegranate aroma	No added aroma	Apple elderflower juice	1–9 rating sweetness	5%
Biggs et al. (2016) [46]	Touch	Extrinsic	Rough plate	Smooth plate	Biscuits	How did the biscuits taste?	Biscuits in smooth plate 3 times more likely to be rated as sweet compared to those in rough plate
van Rompay et al. (2016) [47]	Touch	Extrinsic	Rounded cup surface pattern	Angular cup surface pattern	Hot coffee and chocolate	1–7 rating sweetness	20%
Wang and Spence (2018) [48]	Touch	Extrinsic	Velvet swatch	Sandpaper swatch	Off-dry white wine (10 g/L)	1–9 rating sweetness	13%
Wang and Spence (2018) [48]	Touch	Extrinsic	Velvet swatch	Sandpaper swatch	Fortified red dessert wine (110 g/L)	1–7 rating sweetness	14%

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
