# Peer review of "The Role of Intrinsic and Extrinsic Sensory Factors in Sweetness Perception of Food and Beverages: A Review"

_foods, 2019, doi:10.3390/foods8060211_

Round 1
Reviewer 1 Report
This is an comprehensive review about sweetness enhancement through multisensory interaction as well as interactions between extrinsic and intrinsic factors. Overall, this review will provide academia and industry with valuable information about multifaceted sensory experiences. Still a few modifications are needed for improvement.
1) some references are not cited appropriately. Please check them and link corresponding sentences with them.
2) For oral-somatosensation (lines 259-269),
3) For oral-somatosensation (section 2.5.1), if another aspect - the interaction between flavor components and matrices - can be discussed, it can enrich the review. For instance, release of some volatiles from food matrix is interrupted by not only physical interaction (such as trapping or delayed movement of volatiles in more viscous matrices) but also chemical interaction (hydrophobic bindings, hydrogen bondings, etc). Some readers may be curious whether such interaction soley occurs at the perceptual level or it can occur as a consequence of physicochemical interactions at the food levels. Therefore if the authors can explain who the previous studies on oral-somatosensation can be differentiated from the interaction between food components, it can illustrate the new aspect of flavor perception with regard to the influence of texture more clearly.
Author Response
Response to Reviewer 1
This is an comprehensive review about sweetness enhancement through multisensory interaction as well as interactions between extrinsic and intrinsic factors. Overall, this review will provide academia and industry with valuable information about multifaceted sensory experiences. Still a few modifications are needed for improvement.
1) some references are not cited appropriately. Please check them and link corresponding sentences with them.
2) For oral-somatosensation (lines 259-269),
3) For oral-somatosensation (section 2.5.1), if another aspect - the interaction between flavor components and matrices - can be discussed, it can enrich the review. For instance, release of some volatiles from food matrix is interrupted by not only physical interaction (such as trapping or delayed movement of volatiles in more viscous matrices) but also chemical interaction (hydrophobic bindings, hydrogen bondings, etc). Some readers may be curious whether such interaction soley occurs at the perceptual level or it can occur as a consequence of physicochemical interactions at the food levels. Therefore if the authors can explain who the previous studies on oral-somatosensation can be differentiated from the interaction between food components, it can illustrate the new aspect of flavor perception with regard to the influence of texture more clearly.
We thank the reviewer for their attention and encouraging words
Point 1) We have corrected the source errors on pages 6, 11, and 12.
Point 2) the reviewer did not finish their point so we assume the intended question was addressed in point 3)
Point 3) We have reworked section 2.5.1 to include references on the physical and chemical interactions when it comes to texture and flavor interactions. We have replicated our changes below:
The oral texture or mouthfeel of food and drink can influence the way in which multisensory flavours are experienced (e.g. [119-121]). Christensen [122] first suggested that increased viscosity can reduce taste perception in sweet and salty solutions. In a similar study, this time involving viscosity and flavour intensity, Bult et al. [119] presented a creamy odour either orthonasally or retronasally using an olfactomer; at the same time, milk-like solutions with different viscosities were delivered to the mouth of the participants. The intensity of perceived flavour decreased as the viscosity of the liquid increased, regardless of whether the odour was presented orthonasally or retronasally. Similarly, Weel et al. [121] studied the perception of ethylbutyrate (pineapple like, fruity) and diacetyl (buttery) flavours in whey protein gels, detected by 10 trained panellists. They found that perceived flavour intensity decreased linearly with increasing gel hardness.
Of course, it should be kept in mind that flavour compounds can interact with various ingredients in the food matrix through either chemical or physical interactions, possibly leading to either increased or decreased flavour release (e.g. [123-125]). However, Tournier et al. [126] found that physico-chemical interactions could not explain all the sensory interactions they observed, suggesting that cognitive mechanisms were involved after all.
Reviewer 2 Report
Dear Authors,
I congratulations you of the great article. It is very interesting manuscript.
Critical remarks - Form of Table 1 (table on 6 pages!) is difficult to understand. I propose change Table 1 on horizontally, and decrease font size. On pages 6, 11, 12 are errors of sources.
Author Response
Reviewer 2
Dear Authors,
I congratulations you of the great article. It is very interesting manuscript.
Critical remarks - Form of Table 1 (table on 6 pages!) is difficult to understand. I propose change Table 1 on horizontally, and decrease font size. On pages 6, 11, 12 are errors of sources.
We thank the reviewer for their encouraging words. The final table formatting is of course up to the journal. We have corrected the source errors on pages 6, 11, and 12.
Reviewer 3 Report
Excellent paper, very novel and innovative and perfectly suited for Foods
Author Response
Reviewer 3
Comments and Suggestions for Authors
Excellent paper, very novel and innovative and perfectly suited for Foods
We thank the reviewer for their kind words.